# Effect of Boron in a Hierarchical Nanoporous Layer Formation on Silicate Glass

**DOI:** 10.3390/ma13081817

**Published:** 2020-04-12

**Authors:** Takumi Ito, Erika Tabata, Yuki Ushioda, Takuya Fujima

**Affiliations:** 1Department of Mechanical Engineering, Tokyo City University, Tokyo 158-8557, Japan; g1881005@tcu.ac.jp (T.I.); g1881039@tcu.ac.jp (E.T.); g1981004@tcu.ac.jp (Y.U.); 2Advanced Research Laboratories, Tokyo City University, Tokyo 158-0082, Japan

**Keywords:** silicate glass, hierarchical nanoporous layer, elution, boron, structure formation

## Abstract

A hierarchical nanoporous layer (HNL) can be formed on the silicate glass surface by simple alkali etching. Though it reportedly exhibits various useful functions, such as superhydrophilicity, optical anti-reflection, and material impregnation, the principle of its formation still remains unclear. In this study, HNL formation behavior was experimentally investigated while using scanning electron microscope (SEM) and X-ray photoelectron spectroscopy (XPS) to clarify the role of boron contained in glass. As a result, it was found that HNL formation was significantly promoted by boron, which was rapidly eluted prior to alkali and alkaline earth metals. This suggests that boron, which forms the skeleton structure of glass together with Si and O, elutes to partially decompose the skeleton, and extends the elution route for HNL formation.

## 1. Introduction

Anti-reflection (AR) is one of the functionalities required for practical optical components, such as lenses and filters [1]. Since the reflection of light occurs on the discontinuous surface of the refractive index, the refractive index gap between the air and the optical material is divided into small ones by single or multiple layers with an intermediate refractive index to obtain the AR effect. Therefore, the effect is improved by making the AR film a multilayer film and changing the refractive index in multiple steps, but the manufacturing process is also multi-step. On the other hand, the moth-eye structure [2], with fine and tapered projections, exhibits a continuous change in refractive index within a single structure, thereby realizing the strong suppression of reflectivity.

A hierarchical nanoporous layer (HNL) spatial structure affects the same refractive index change to the moth-eye structure [3]. HNL is a porous structure formed on the glass surface in which the pore diameter gradually decreases from the apparent surface toward the bulk region, whereby the refractive index continuously increases. This structure also exhibits long-life super-hydrophilicity and various functionalities associated therewith [4], and a material impregnation property [5], as well as the above-mentioned AR.

Another feature of HNL is a remarkable easiness on its structure formation. HNL with the above properties can be formed on the silicate glass surface by one-pot etching, but only by heating the silicate glass in an aqueous solution of sodium hydrogen carbonate. However, the principle by which such complicated and ordered mesoscale structure is formed (only by etching) has not been elucidated.

A reaction under similar conditions is also likely to occur when groundwater comes into contact with nuclear waste glasses that are buried deep underground to sequester nuclear waste [6]. Borosilicate glass is most widely used to immobilize nuclear waste by melting and vitrificating the waste and glass together. Since groundwater can have an alkaline pH [7], its contact with vitrified borosilicate glass is similar to the HNL-forming condition. Therefore, it is useful to understand the HNL structure’s formation principle when examining the corrosion resistance of vitrified materials that are stored underground for a long time.

In general applications, such as popular bottles and windows, soda-lime glass is used among silicate glasses, but borosilicate glass is used in applications that require higher hardness and heat resistance. Borosilicate glass has a small wavelength dispersion of the refractive index and it is often used for optical components. These properties are derived from the composition of borosilicate glass, which contains more boron and less sodium and calcium than soda-lime glass [8].

According to Zachariasen’s model [9], the basic unit in silicate glass is a SiO_4_ tetrahedron with four oxygen peaks around Si, and the units are linked by an oxygen. Oxygen atoms that are shared by two tetrahedrons are called bridging oxygen (BO), and oxygen that is not shared and is bound only to a single tetrahedron is called non-bridging oxygen (NBO). The NBO forms a random, three-dimensional network structure due to the discontinuity in the tetrahedron connection. The free space that is generated near the NBO due to the absence of covalent bonds is occupied by the alkali or alkaline earth metal contained in the silicate glass [10]. Regarding the cationic elements in the molecular structure of such a glass, the elements that constitute a three-dimensional network by covalent bonds (Si, B, P, etc.) and elements that occupy free space by NBO (Na, K, Ca, etc.) are categorized as network formers (NWF) and network modifiers (NWM), respectively [11,12].

Various elements dissolve out of silicate glass upon contact with water or aqueous solution, and the chemical composition of silicate glass affects the elution behavior [13]. Si, B, O, and Na are the main components contained in borosilicate glass. A Si-O-B fragmentation rate is reportedly higher than that of Si-O-Si in the reaction with water [14]. That is, it is considered that HNL formation is accelerated in a glass containing boron, such as borosilicate glass. In this work, we investigated the effect of B in glass on HNL formation rate with composition-controlled glass.

## 2. Materials and Methods 

In order to investigate glass with and without boron, we made one based on Si-B-Na-Ca. Since the glass could not be made stably with a composition that excludes boron from the borosilicate glass, calcium was also added for the stabilization. We mixed powders of SiO_2_ (Guaranteed Reagent), B_2_O_3_ (Reagent, >90%), Na_2_CO_3_ (Guaranteed Reagent, >99.8%), and CaCO_3_ (Guaranteed Reagent, >99.5%) under a molar ratio of Si:B:Na:Ca as 7:x:2:1 (x = 0.0, 2.0). Next, we melted it in a platinum crucible at 1600 °C for 30 min., and then quenched it in a graphite mold to vitrify. We purchased all of the raw powders from FUJIFILM Wako Pure Chemical Corp. (Osaka, Japan) and used them without further purification. 

The obtained rod-shaped glass had a diameter of 15 mm. It was sliced to a thickness of 1.3 ± 0.5 mm by a rotary grindstone cutter (HS-45AC, Heiwa Technica Co., Ltd., Kanagawa, Japan) and then polished with cerium oxide, using an auto-polisher (Rana-3, IMT Co., Ltd., Wakayama, Japan), until the optical unevenness disappeared. The molecular structure in the obtained glasses was characterised by a Raman spectrometer (T64000, HORIBA, Ltd., Kyoto, Japan) while using an Ar laser (514.5 nm, 2 mW).

The HNL formation on the obtained glass materials was performed by keeping the glass materials in a 0.5 mol/L of sodium hydrogen carbonate aqueous solution at 100 °C. 

The processed sample was divided, and the cross-section and surface were observed while using a scanning electron microscope (SEM, SU8230, Hitachi High-Technologies, Hitachi, Japan). For insulating materials, like glass, a conductive coating, such as platinum palladium, is used as a pretreatment in usual electron microscopy observations, but, in this study, the structure to be observed is so small that it could be buried under the coating. Therefore, in this study, the measurement was performed without any coating, but with a conducting carbon tape around the observation field. A secondary electron image was acquired for a primary electron acceleration voltage of 1.0 kV.

We used X-ray photoelectron spectroscopy (XPS, SSX-100, Surface Science Instruments, Mountain View, CA, USA) with Al Kα ray to investigate the composition and chemical bonding state of the treated samples. A Nickel mesh on the samples and a flood gun suppressed charge-up on the insulating samples. The obtained data’s binding energy was corrected, so that the C 1s spectrum of the contamination by hydrocarbons on the sample surface became 285 eV. 

## 3. Results and Discussion

Figure 1 shows the Raman spectra of the prepared glasses with x = 0.0 and 2.0. The peaks around 950 and 1100 cm^−1^ are derived from the Si-O stretching vibration of Q^2^, the tetrahedral unit with two BOs, and Q^3^, with three BOs, among the four oxygens, respectively [14,15]. The Q^3^ peak shifted to lower energies with the boron content, which indicated an increase of the overall bond length in the glass to relax the compression or exert the tensile stress. This shift is considered to arise from the residual stress difference due to the thermal history during glass production. The peak of Q^2^ for x = 2.0 was smaller than the other. This suggests that NBO was less likely to be formed in x = 2.0, because the amount of Na, an NWM, was relatively small.

Figure 2a shows a typical surface of the HNL, x = 2 a sample after 16 h treatment, with a porous size of few tens of nm on the surface. Figure 2b shows cross-sectional SEM images near the surface of both samples at x = 0.0 and 2.0 for various treatment times. The HNL structure did not appear after the 8 h treatment for x = 0.0, and that had a thickness less than 200 nm for 20 h. On the other hand, ca. 700 nm of HNL was formed for x = 2.0 even in 8 h. Thus, a significant difference was found in the rate of HNL formation by the content of boron.

Figure 2c shows the dependence of the thickness of the HNL structure on the HNL formation processing time in glasses of both compositions based on the SEM observation. The distinct thicknesses shown in Figure 2b are discussed here, as the evaluation does not include pores that are too small to be observed by SEM. This Figure shows that HNL formation progresses with processing time rather than at a constant rate, but with an exponential manner. Additionally, the glass containing boron started the HNL formation almost without any threshold regarding the processing time, whereas that without boron had one of about 10 h, during which the structure was not formed, which suggested that a reaction for the HNL formation had been suspended.

Figure 3a shows an HNL formation treatment time dependence of the chemical composition of the sample surface based on XPS for the composition of x = 2.0. While the ratios of Si and O did not change significantly with the treatment time, B, Ca, and Na continuously decreased. This indicates that HNL is formed by elution of elements from the pristine glass, other than any deposition on the glass.

The elution rate of boron is higher than that of alkali and alkaline earth metals, as shown in the Figure. Furthermore, as the structure formation progressed, the eluting elements fell below the detection limit, and the composition gradually changed to a binary only of Si and O. As shown in Figure 3b, the ratio of O to Si in sample surface, the HNL, also approached that of quartz, 2, as the etching proceeded.

Figure 4 shows the O 1s spectra of the XPS used in Figure 3. The spectra were fitted with the Voigt function after subtracting the background with the Shirley method [16]. The main peak component, as indicated by the red line, and the small one, as indicated by the blue line at lower binding energy, are attributed to BO and NBO, respectively [17]. As shown in the Figure, the peak intensity of NBO decreased after a 5 h treatment in comparison with the untreated glass and disappeared after 20 h. The result of this spectral composition is consistent with that for the chemical composition that is shown in Figure 3.

Boron in the glass remarkably increased the rapidity of HNL formation, according to Figure 2. In understanding the mesoscopic phenomenon, it should be particularly noted that, from a microscopic point of view, the elution rate of boron is high in the HNL formation process, and NWM followed, as shown in Figure 3a. Though NWM in glass is known to elute from the interspace near the NBO upon contact with water, in this work, the NWF boron did so at a higher rate by partially decomposing the network of tetrahedral units. It is thought that this expanded the route for NWM elution and promoted the overall elution and HNL formation.

## 4. Conclusions

In this study, we investigated the effect of boron on the formation of a hierarchical nanoporous layer (HNL) by forming it while using glasses with different compositions. The thickness of the layer of pores with the SEM-observable size revealed that HNL formed on a glass containing boron significantly faster than on one without boron. XPS analysis for the sample surface revealed that boron eluted rapidly with the treatment, followed by Na and Ca, and the composition of HNL finally approached that of quartz. It is thought that the elution of boron, which takes part in the network skeleton of glass, expanded the elution path of the elements and significantly promoted HNL formation. Therefore, the presence of boron in glass is a key rate factor in the formation of HNL.

## Figures and Tables

**Figure 1 materials-13-01817-f001:**
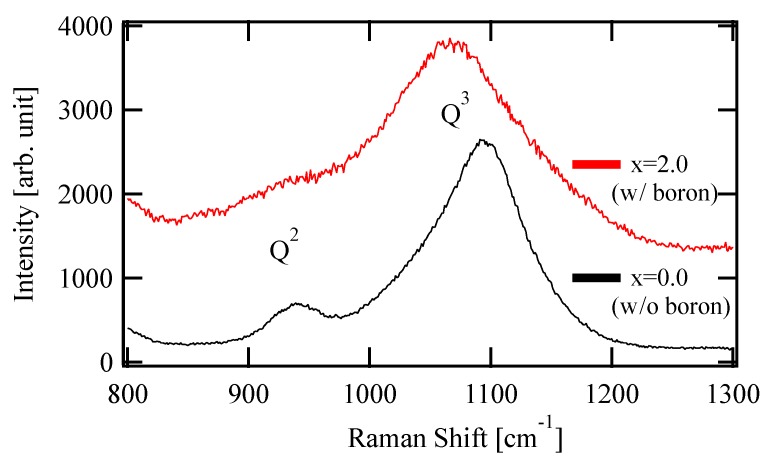
Raman spectra of composition-controlled glass (x = 0.0 and 2.0)

**Figure 2 materials-13-01817-f002:**
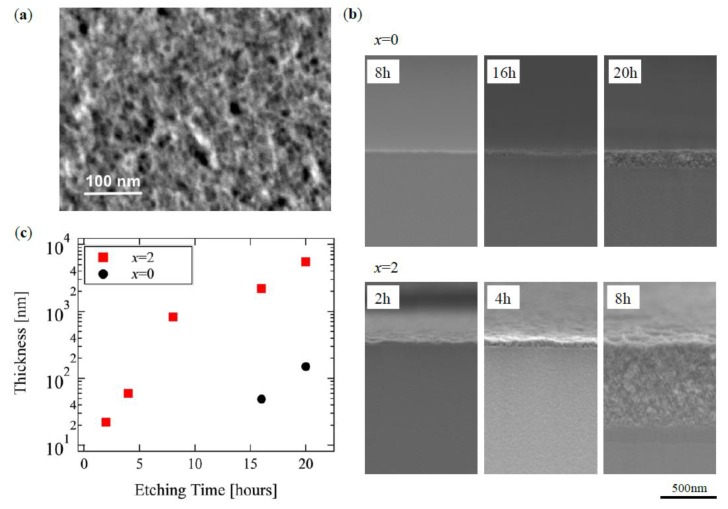
Scanning electron microscope (SEM) micrographs of (**a**) surface of x = 2 sample after 16 h treatment and (**b**) cross-section of both chemical composition of glass (x = 0.0, 2.0). The hierarchical nanoporous layer (HNL) thickness is shown in (**c**) as a function of the etching time.

**Figure 3 materials-13-01817-f003:**
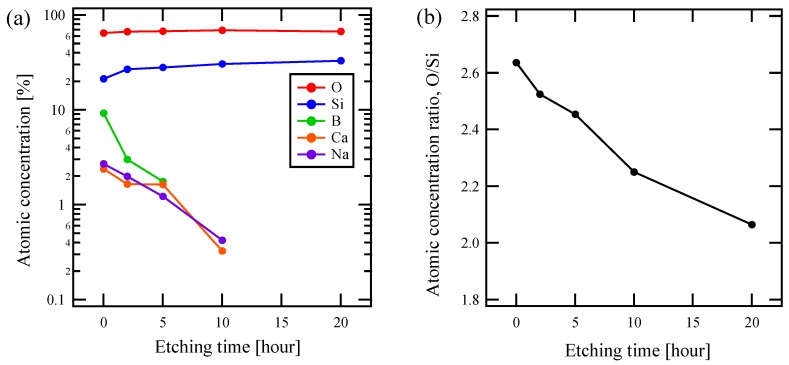
Chemical composition of x = 2.0 glass by X-ray photoelectron spectroscopy (XPS) measurements as a function of etching time about (**a**) atomic fraction of each element contained in glass and (**b**) atomic fraction ratio of O/Si.

**Figure 4 materials-13-01817-f004:**
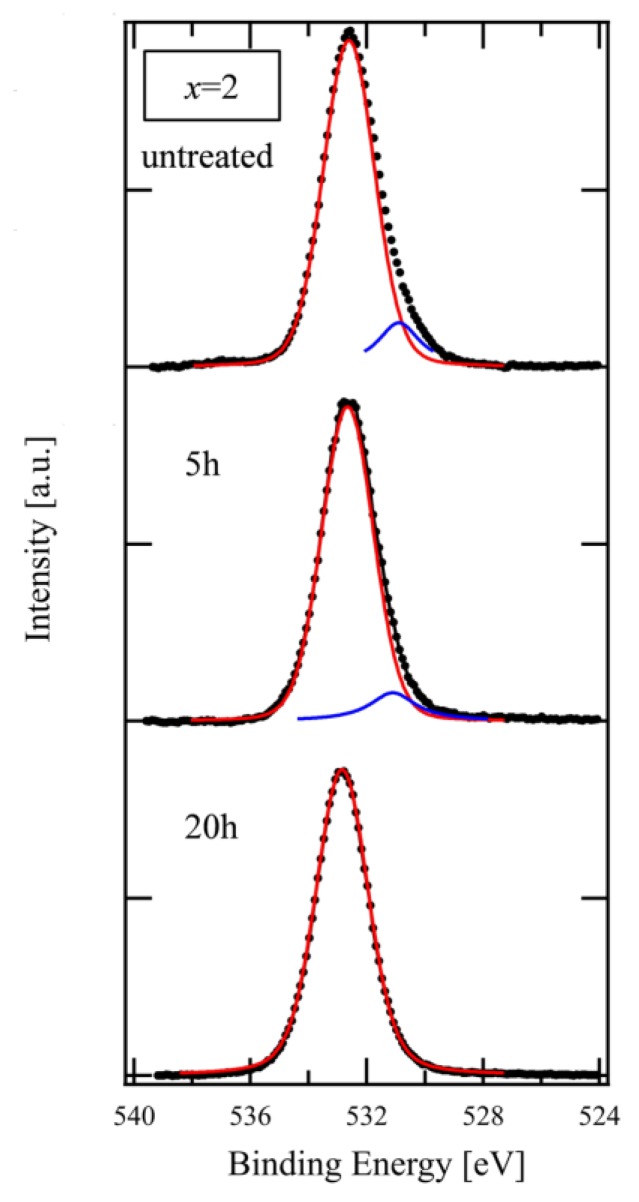
XPS spectra of O 1s on glass of x = 2.0 composition for different etching time (black dots). Red and Blue lines represent BO and NBO components, respectively, decomposed from the experimental data using Voigt function.

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
