# Peer review of "Effect of Boron in a Hierarchical Nanoporous Layer Formation on Silicate Glass"

_materials, 2020, doi:10.3390/ma13081817_

Round 1

Reviewer 1 Report

The manuscript reports on the formation of hierarchical nanoporous layers in glasses of different composition by etching in NaHCO3 over time. The study on itself is interesting and the manuscript is well written, however, there are some minor points to be considered. Since authors discussing hierarchical nanoporosity of the glasses, the corresponding techniques for proving these findings should be applied. For instance nitrogen physisorption at 77 K and subsequent applying of corresponding models to prove hierarchical nanoporosity of the glasses.

Author Response

   The reviewer evaluated the study as interesting, and proposed some experiment to prove the porous structure of HNL, such as the low-temperature nitrogen adsorption method as a minor improvement.

   The formation of a hierarchical nanoporous layer by the etching process described in the manuscript has already been established in the authors' previous reports. The hierarchy, therein, was evaluated by the SEM micrography on the structural cross-section, as in this study. In any case, this study focuses on the rate of HNL formation, not on the degree of hierarchy.
   The low-temperature nitrogen adsorption suggested by the reviewer is certainly one applicable technique, but as the reviewers have mentioned, it depends on the model in interpreting the data. The authors would like to consider the future application of the method as a supportive method for evaluation by SEM though not having the equipment at this time.

Reviewer 2 Report

The manuscript by Ito et al. deals with the effect of boron on the formation of hierarchical nanoporous layer on silicate glass. It is a quite short article that could be enriched with further experimental techniques and discussion. For example, the main claim of the manuscript regarding the boron effect on the layer formation is obviously supported by the SEM micrographs, but the overall given information is close to minimum. Dealing with a porous material, the reader would expect to see at least a pore size or a distribution, as well further information if possible as surface area, pore volume, etc. I understand that the latter is rather impossible for a porous layer of a few hundred nm, however, a higher magnification (with SEM if possible) of the surface or the cross section would give more information about the pores, and not just the thickness of the porous layer. Besides, the layer is said to be hierarchical because we know that it would be that kind, but this is not supported by any experimental data. In addition, the Q3 Si-O Raman peak is shifted by ca. 20-30 cm-1 when boron is added but there is no explanation. The SEM micrographs for the x=2 glass seem to be more magnified than the x=0 glass as the surface details seem to be more discreet. Are all the micrographs of the same magnification? XPS survey spectra could be presented as supporting information. Figure 3, correct etching.

Author Response

   The reviewer suggested that more experimental data and discussion could enrich this manuscript. The authors, therefore, reinforced the manuscript following the reviewer's advice. The point-by-point response for each topic is as follows.

> "a higher magnification (with SEM if possible) of the surface or the cross section would give more information about the pores, and not just the thickness of the porous layer."

   The authors agree with the reviewer's point, and have added a surface SEM micrograph in the fig. 2. We also considered increasing the magnification of the SEM micrographs of the cross-section, but did not. It was difficult to obtain a higher resolution because the samples, insulating glass with nano-structure, should be observed without conductive coating by SEM to occur charging up easily.

> "Besides, the layer is said to be hierarchical because we know that it would be that kind, but this is not supported by any experimental data."

   At this stage, the authors have not been able to evaluate the hierarchy quantitatively, and have already confirmed it in SEM images of the structural cross-section. Currently, we are still in progress to study a quantitative evaluation method of hierarchy. So in this study, the authors focus on the HNL formation speed, not on the degree of hierarchy.

> "In addition, the Q3 Si-O Raman peak is shifted by ca. 20-30 cm-1 when boron is added but there is no explanation."

   As the reviewer pointed out, the discussion on the Raman spectra was only about Q2 that the authors were interested in. Discussion about Q3 has been added in the revised manuscript, which does not change the total conclusion.

> "The SEM micrographs for the x=2 glass seem to be more magnified than the x=0 glass as the surface details seem to be more discreet. Are all the micrographs of the same magnification?"

   All SEM images of the sample cross-section are of the same scale and are consistent with fig.2 (b). The difference in the appearance of the surface is due to the sample installation angle during observation.

> "XPS survey spectra could be presented as supporting information."

   In the XPS survey data, the peak of boron does not appear clearly due to its low detection sensitivity, causing misunderstanding about the chemical composition. Therefore, the authors will refrain from posting the survey data for this time.

> "Figure 3, correct etching."

   The authors thank the reviewers for their comments and have corrected them.

Reviewer 3 Report

The manuscript entitled “Effect of boron in a hierarchical nanoporous layer formation on silicate glass” focuses on the study of the effect of boron on the formation of a hierarchical nanoporous layer on the silicate glass surface.

I recommend this paper for publication in the Materials, but I think that some parts of it need improvement. In my opinion:

- in the introduction, the aim and subject of the study is very briefly written (lines 65-66),

-SEM measurement conditions are not described in the experimental section.

-Raman Spectra analysis is not very accurate.

Author Response

   The reviewers recommended publishing the authors' manuscript in "materials" and suggested several improvements.

> "in the introduction, the aim and subject of the study is very briefly written"

   As the reviewer pointed out, the aim and subject were too short, so the authors have added some detailed description.

> "SEM measurement conditions are not described in the experimental section."

   This was also a lack of explanation by the authors. More detailed measurement conditions have been added to "Materials and Methods."

> "Raman Spectra analysis is not very accurate."

   Raman scattering is a very widely used measurement technique for understanding bonding structures in the glass. However, since the Raman shift amount easily changes due to the residual stress in the material, the analysis by this method may be "not very inaccurate." Although the characteristics of the data coming from the measurement principle cannot be changed, the revised manuscript provides an enriched discussion about the Raman spectra.

Round 2

Reviewer 1 Report

Authors replies on the comments and pointed out on the qualitative study of nanoporous structure of glasses. Using SEM technique only macroporosity can be addressed qualitatively, but no micro- and mesopores. Utilization of physisorption techniques access all pores, from micro to macropores. Authors have to insert the clear statement in discussion and conclusion sections that only macroporosity is analysed in the manuscript.

Author Response

Following the reviewer's request, authors have inserted the clear statement in discussion and conclusion sections that only macroporosity is analysed in the manuscript.

Reviewer 2 Report

The manuscript would be even better with further characterization but this is not possible due to lack of proper instrumentation or the nature of the materials. Nevertheless, authors have made appropriate clarifications and modifications. Hence, I recommend publication as it is.

Author Response

The authors appreciate the reviewer's recommendation to publish the manuscript as it is in "materials".

Reviewer 3 Report

The manuscript was revised according to my recommendations. In the current form it can be published in the Materials.

Author Response

The authors appreciate the reviewer's decision that allow the manuscript published as it is in "materials".